# Advanced Pancreatic Ductal Adenocarcinoma: Moving Forward

**DOI:** 10.3390/cancers12071955

**Published:** 2020-07-18

**Authors:** Caspar Franck, Christian Müller, Rosa Rosania, Roland S. Croner, Maciej Pech, Marino Venerito

**Affiliations:** 1Department of Gastroenterology, Hepatology and Infectious Diseases, Otto-von-Guericke University Hospital, 39120 Magdeburg, Germany; caspar.franck@med.ovgu.de (C.F.); christian.mueller@med.ovgu.de (C.M.); rosa.rosania@med.ovgu.de (R.R.); 2Department of General-, Visceral-, Vascular- and Transplant Surgery, Otto-von-Guericke University Hospital Magdeburg, 39120 Magdeburg, Germany; roland.croner@med.ovgu.de; 3Department of Radiology and Nuclear Medicine, Otto-von-Guericke University Hospital, 39120 Magdeburg, Germany; maciej.pech@med.ovgu.de

**Keywords:** pancreatic ductal adenocarcinoma, palliative therapy, maintenance therapy, supportive therapy, biomarker, molecular subtypes, pancreatic enzyme replacement, prophylaxis of thromboembolisms, small intestinal bowel obstruction, staging

## Abstract

Globally, the death rate of pancreatic ductal adenocarcinoma (PDAC) has doubled over 30 years and is likely to further increase, making PDAC a leading cause of cancer-related death in the coming years. PDAC is typically diagnosed at an advanced stage, and modified FOLFIRINOX or nab-paclitaxel and gemcitabine are the mainstay of systemic therapy. For elderly patients with good performance status, low-dose treatment can preserve quality of life without compromising cancer control or survival. Maintenance therapy should be considered in PDAC patients achieving disease control with systemic therapy. In particular, olaparib has demonstrated a progression-free survival benefit of 3.6 months in a subgroup of PDAC patients with germline BRCA1/2 mutations (ca. 10% of all PDAC). Pancreatic enzyme replacement therapy is often omitted in the treatment of patients with PDAC, with possibly deleterious consequences. Small intestinal bacterial overgrowth is highly prevalent in patients with PDAC and should be considered in the diagnostic algorithm of PDAC patients with bloating and diarrhea. Rivaroxaban has been associated with a reduced risk of thrombosis without an increase in major bleeding events, and its use should be considered in every patient with advanced PDAC undergoing systemic therapy.

## 1. Introduction

Pancreatic ductal adenocarcinoma (PDAC) is a substantial public health concern and socioeconomic burden. With an average loss in life expectancy of 14.7 years and a severe reduction of life quality for the sufferers, PDAC is responsible for the loss of nearly one million quality-adjusted life-years (QALYs) per year alone in Europe [1]. According to data from the World Health Organization (WHO), in 2018, worldwide for both sexes, PDAC was the twelfth most common cancer (458,918 incident cases) and the seventh leading cause of cancer-related death (432,242 deaths) [2]. In Europe and in the USA with an estimated 128,045 and 45,574 deaths in 2018, respectively, the disease ranks as the fourth and the third leading cause of cancer-related death. PDAC incidence is greater in males than in females (European age-standardized annual incidence rate 9.3/100,000 vs. 6.3/100,000) with a male-to-female ratio of 1.5:1 [2]. Most PDAC (about 90%) are considered sporadic, whereas inherited PDAC are less common (ca. 10%). In general, the age-standardized incidence of PDAC is higher in the industrialized parts of the world. However, the survival rates are similar among high-income, middle-income, and low-income countries [3]. Due to a further increase in the incidence, PDAC has been projected, for 2030, to become the second leading cause of cancer-related death in the USA and in Germany [4,5]. For 2040, the WHO has predicted 37,855 (29.3%) additional cases yearly, in Europe only, nearly completely in the age group ≥70 years on the basis of an ongoing ageing of the population [2]. PDAC is rare before the age of 30 years. However, after 30–35 years the incidence rises exponentially [6,7]. The nearly equal incidence and mortality rate indicate the very poor prognosis of PDAC with still today virtually all sufferers dying from the disease. For PDAC arising within intraductal papillary mucinous neoplasm (IPMN), the prognosis is more favorable with a five-year survival reported between 40% and 60%. However, this is due to comparatively lower tumor stages and better pathologic characteristics at the time of surgery. For advanced IPMN-associated PDAC, the prognosis is comparable to non-IPMN-associated PDAC [8]. In the U.S. “Cancer Statistics, 2019”, among the investigated malignancies, the five-year relative survival rate for PDAC was the lowest, with 9% for all stages and only 3% for metastatic disease, which is mostly the case [9]. In Germany, besides mesothelioma, the survival rate is also the lowest among all types of cancer [7]. Remarkably, although the survival rates for most cancer diseases have clearly improved over the past decades, advancements in survival for PDAC have been limited. In the Global Burden of Disease Study, smoking, diabetes, and obesity were responsible for 21.1%, 8.9%, and 6.2% of PDAC deaths, respectively [3]. These are all modifiable risk factors, and thus implementation of prevention strategies is warranted. 

## 2. Methods

The authors performed an independent search in PubMed, on May 2020, for publications on PDAC. There was no filter applied to the searches. The Boolean operator “AND” was used to narrow the search results. The following search term combinations were used: “pancreatic ductal adenocarcinoma” (all fields) AND "epidemiology" (all fields), “pancreatic ductal adenocarcinoma” (all fields) AND "systemic therapy" (all fields), and “pancreatic ductal adenocarcinoma” (all fields) AND "supportive therapy" (all fields). Studies included and discussed in the present review, were selected by the authors for their relevance and importance to the field of PDAC.

## 3. Molecular Subtypes

Several subtyping systems for PDAC have been proposed. Table 1 shows, for example, the subtyping system proposed by Bailey et al., which is based on a whole-genome and deep-exome sequencing, with gene copy number analysis of 456 PDAC samples [10].

The classification took into account the most common driver mutations observed in the PDAC. For example, KRAS mutations are present in about 90% of all PDAC samples and are linked to many intracellular signaling pathways [11,12]. 

The squamous subtype showed the most heterogenic gene programs involved in tumor progression and was associated with altered transcription in p63, hypoxia-related genes, TGFβ, MYC, WNT, and autophagy related genes. 

The immunogenic subtype with upregulation of CTLA4 and PD1 are mostly involved in 29% of the analyzed samples.

The aberrantly differentiated endocrine exocrine (ADEX) subtype with 21% incidence in the samples, was characterized by alterations in exocrine function and secretion, involving MIST1, NR5A2, MODY, and others. These genes are upregulated by mutant KRAS expression. 

The less common pancreatic progenitor subtype accounts for 19% of the samples, includes alternations in several transcription factors (PDX1, HNFS, FOXAS, and HES1), and metabolic features (oxidative enzymes, glycosylation, and drug metabolism). 

From the clinical point of view, this classification offers some prognostic information, but has unfortunately no therapeutic implications.

Recently, three subtyping systems were harmonized to reveal two distinct PDAC subtypes with significant implications for prognosis and treatment response to FOLIFIRNOX [8]. Although, this subtyping system clearly represents a step forward in the management of PDAC, it still leaves many clinical relevant questions unanswered, such as the following: (1) which regimen to offer to patients with anticipated resistance to FOLIFIRNOX, (2) which therapy to offer in case of resistance to first line therapy, and (3) the prediction of early relapse after surgery.

## 4. Mainstay (First, Second Line, and Biomarker Driven Treatments)

### 4.1. First-Line Treatments

Surgical resection is still the only potentially curative option for the minority of patients (10–20%) with operable PDAC. However, the vast majority of patients with PDAC undergoing surgical resection, experiences disease recurrence [13]. For PDAC patients with metastatic disease or with locally advanced disease not amenable to surgery, systemic therapy can be offered to prolong life expectancy and improve or preserve quality of life. In the palliative setting, physicians´ commitment is crucial in balancing survival advantages of a systemic therapy with its potential adverse events and possible negative impact on quality of life. The Eastern Cooperative Oncology Group (ECOG) performance status helps in selecting PDAC patients that could benefit from a systemic therapy. Conceivably, for patients with poor performance status (ECOG 3/4) or poorly controlled comorbidities, supportive treatment is the standard of care, as the harm from a tumor-specific therapy outweighs potential benefits in this specific population [14,15,16]. 

In 1996, gemcitabine was approved for PDAC not amenable for surgery. In a randomized trial, 23.8% of patients receiving gemcitabine monotherapy experienced a clinical benefit response (defined as sustained improvement of pain, Karnofsky performance status, or weight) as compared with 4.8% of the patients treated with 5-fluorouracil (*p* = 0.002) [17]. There was also a modest, but significant difference in median overall survival (OS 5.65 vs. 4.41 months, *p* = 0.003) in favor to gemcitabine.

Since the results of the investigator initiated ACCORD 11 trial have been presented in 2011, FOLFIRINOX (oxaliplatin 85 mg/m^2^ d1, irinotecan 180 mg/m^2^, leucovorin 400 mg/m^2^, 5-fluorouracil 400 mg/m^2^, 5-fluorouracil 2400 mg/m^2^ 46 h, and qd15) has become a new standard of care in the first-line therapy of patients with metastatic PDAC and good performance status (ECOG score 0/1) [14,16,18]. FOLFIRINOX offered a survival benefit of 4.3 months over gemcitabine monotherapy in patients with metastatic PDAC (median OS 11.1 vs. 6.8 months, *p* < 0.001) [19]. Despite the high rates of grade 3/4 neutropenia, febrile neutropenia, thrombocytopenia, diarrhea, and sensory neuropathy, patients receiving FOLFIRINOX experienced a significant delay in deterioration of life quality as compared with patients treated with gemcitabine (degradation in quality of life at six months of treatment in 31% vs. 66%). FOLFIRINOX is a standard of care option in the treatment of patients with metastatic PDAC, although no approval from regulatory agencies exist (off-label use). In clinical practice, many oncologists reduce the dose of one or more of the FOLFIRINOX components (5-FU, oxaliplatin, or irinotecan) by 15% to 50% in order to reduce therapy-associated toxicity. These dose adjustments vary slightly from study to study or clinic to clinic but altogether are summarized under the term modified (m)FOLFIRINOX. Although a direct, prospective, randomized comparison of the regimens is missing, mFOLFIRINOX was shown to offer a comparable level of efficacy with a lower rate of adverse events, and thus was equivalent to FOLFIRINOX implemented into current guidelines for palliative treatment of PDAC [18,20,21,22]. Table 2 shows dosages and side effects of the different FOLFIRINOX formulations investigated in phase III trials [19,23].

In 2013, the doublet gemcitabine/nab-paclitaxel (GA) was approved as a first-line treatment for advanced PDAC. In the MPACT trial, the combination proved to have a significant survival benefit (OS 8.5 vs. 6.7 months, *p* < 0.001) for metastatic PDAC [24] as compared with solely gemcitabine. In the GA arm, the safety profile was inferior to gemcitabine monotherapy; grade 3/4 neutropenia, leucopenia, fatigue, and peripheral neuropathy were more common. Interestingly, nab-paclitaxel-induced neuropathy was reversible after withdrawal of the drug in most patients (median time to improvement from grade 3 to grade 1 or resolution of the event in 29 days) and resuming the treatment with dose reduction was often possible. It should be noted that in the MPACT trial, patients with reduced performance status (8% ECOG score 2) and patients ≥75 years (10%) were also included, which could, in part, explain the lower survival benefit in contrast to the FOLFIRINOX trial (1.8 vs. 4.3 months). A limitation of the MPACT trial is that no assessment of life quality was performed. 

FOLFIRINOX and GA both represent well-established, largely equivalent first-line regimens for patients with advanced PDAC in good performance status (ECOG ≤ 2). However, no randomized trials comparing the two schemes head-to-head or investigating their optimal sequence of application have been performed. Moreover, no predictive marker is available to guide decision making in selecting one of the two regimens. In 2019, a meta-analysis evaluating 16 retrospective studies (3813 patients) comparing both regimens, proved to have a small advantage for FOLFIRINOX in terms of median OS (1.15 months), however, the progression-free survival (PFS) and overall response rate were similar [25]. In this meta-analysis, among serious toxicities (grade 3/4), neutropenia, febrile neutropenia, and nausea were found to be lower with GA, whereas neurotoxicity and anemia were lower with FOLFIRINOX. The frequency of dose reductions and discontinuations did not significantly differ between the two regimens.

In the case of reduced performance status (ECOG 2), current NCCN guidelines have proposed GA for selected patients [16,18]. Alternatively, in this group, gemcitabine monotherapy has been endorsed, which could also be a treatment possibility in patients with serum bilirubin level >1.5×, the upper limit of normal (ULN). On the basis of a phase II trial, patients with serum bilirubin >1.5× ULN have also been offered FOLFOX [26,27].

BRCA1/2- (BReast CAncer 1/2) or PALB2 (Partner and Localizer of BRCA2) gene mutations are associated with a defective DNA damage response, which increases sensitivity to DNA damaging agents such as platinum agents. Recently, treatment with gemcitabine/cisplatin (GC) for the small proportion of germline BRCA1/2- or PALB2-mutated PDAC (5–9%), was shown to be highly effective in a phase 2 study, whereas the concurrent addition of the polyadenosine diphosphate-ribose polymerase (PARP) inhibitor veliparib did not further improve efficacy [28]. First-line treatment with GC for inoperable BRCA1/2- or PALB2-mutated PDAC resulted in a median PFS of 9.7 months and a median OS of 16.4 months, respectively. The two- and three-year survival rates for the entire cohort were 31 and 18 percent, respectively, which represent among the longest reported in any randomized trial in PDAC. Although there was no standard control arm (i.e., FOLFIRINOX), the authors concluded that their data established GC as a standard-of-care for germline BRCA1/2- or PALB2-mutated PDAC. Current NCCN guidelines endorse GC (in addition to FOLFIRINOX) as first-line treatment for BRCA1/2- or PALB2-mutated PDAC [18]. Notably, although data for somatic BRCA1/2- or PALB2-mutated PDAC is missing, this treatment is suggested for patients with either germline or somatic mutations.

Consequently, the choice of first-line chemotherapy depends on individual experiences, local standards, patient’s comorbidities, the toxicity profile, and innovatively, the mutational status of the tumor.

Real world data from Europe, collected by 225 physicians from 2565 patients, treated in 2014 to 2016, yielded FOLFIRINOX (35.6%), GA (25.7%), and gemcitabine monotherapy (20.5%) as the most commonly used first-line regimens [29]. Interestingly, in France and the UK, FOLFIRINOX was prescribed more frequently than GA, whereas in Italy and Spain, GA was more often selected for first-line therapy. 

### 4.2. Second-Line Treatments

The choice of the second-line regimen is based on first-line treatment. In the event of a prior gemcitabine-based therapy, the NAPOLI regimen is approved for second-line treatment [30]. In the randomized phase 3 NAPOLI-1 trial, patients with metastatic PDAC (ECOG 0–2) previously treated with gemcitabine-based therapy where randomized to receive either nanoliposomal irinotecan (nal-IRI) plus 5-FU/leucovorin, 5-FU/leucovorin, or nal-IRI alone. Patients assigned to the combination arm had a survival benefit of 1.9 months as compared with 5-FU/leucovorin with amedian OS of 6.1 months (95% CI 4.8–8.9) vs. 4.2 months (95% CI 3.3–5.3), respectively (*p* = 0.012). Nal-IRI alone (OS 4.9 months, 95% CI 4.2–5.6) showed no benefit as compared with 5-FU/leucovorin. Other second-line therapy options after GA, are FOLFIRINOX/mFOLFIRINOX and FOLFIRI, especially where Nal-IRI is not available. A recently published retrospective study showed, for patients in good performance status (ECOG 0/1) with progressing disease on GA, showed a median OS of 7.0 months under a second-line treatment with mFOLFIRINOX (omission of 5-FU bolus) [31]. The median OS after the start of first-line therapy was 15.9 months. Current NCCN Guidelines suggest second-line therapy with mFOLFIRINOX after failure of GA for patients in still good performance status (ECOG 0/1) [18].

With respect to patients with progressive disease after a fluoropyrimidine-based therapy (i.e., FOLFIRINOX), no approved regimens are available. However, in clinical practice, because of the lack of further options in this indication, for patients who continue to have a good or acceptable performance status (ECOG 0–2), gemcitabine/nab-paclitaxel is often offered as a second-line treatment. This approach is also recommended by current guidelines [14,18]. In a French prospective observational study with 57 metastatic PDAC patients (median age 59.9 years, ECOG 0–2) treated with GA after failure to FOLFIRINOX, the median OS under second-line therapy was 8.8 months, and it was 18 months after the start of first-line FOLFIRINOX. Serious side effects (grade 3/4) occurred in 40% of the patients [32]. Notably, a retrospective Austrian study comparing both administration sequences (1st-line FOLFIRINOX and 2nd-line GA vs. 1st-line GA and 2nd-line FOLFIRINOX) revealed no significant difference with respect to OS and PFS [33].

Gemcitabine monotherapy, which is widely used for second-line treatment in patients with reduced general condition (ECOG 2), has been shown to be the most commonly used second-line regimen (45.9%) in Europe after failure of FOLFIRINOX.

Until today, because of the rapidity of progression in most of all patients, for PDAC there is no data available to recommend any third-line therapy.

### 4.3. Biomarker-Driven Treatments

Mismatch repair deficiency (dMMR/MSI-H) is found with a low frequency of about 1% in PDAC [34]. Even if available data for immunotherapy for dMMR PDAC is rare, efficacy of pembrolizumab generally seems to be discouraging. In the phase 2 KEYNOTE-158 study, the median PFS and OS in a cohort of 22 previously treated PDAC patients was only 2.1 and 4.0 months, respectively [35]. However, four objective responses (18%), one of which was complete, were observed. Current guidelines recommend routine testing of MMR in advanced PDAC and, consecutively, second-line treatment with pembrolizumab [14]. 

For germline BRCA1/2- or PALB2-mutated PDAC previously treated with FOLFIRINOX, GC rather than GA can be a reasonable second-line regimen [18]. 

Less than 5% of PDAC shows neurotrophic receptor-tyrosine-kinase (NTRK) gene fusions. NTRK inhibitors (i.e., Larotrectinib) have received approval for treating any tumor with evidence of an NTRK fusion, irrespective of the primary tumor location (tissue-agnostic indication) [36]. Fusion of TRK-coding genes (NTRK fusion) leads to a constitutive activation of several downstream signaling pathways, including RAS-MAPK (proliferation), PI3K-AKT (survival/anti-apoptosis), or PLCγ-PKC (differentiation). In a case report, Larotrectinib was well tolerated and improved quality of life for a patient with PDAC harboring an NTRK1 gene fusion. However, some patients can develop resistance to therapy [37]. To overcome this resistance, a next-generation TRK inhibitor is currently in development.

In a retrospective analysis of 1082 samples of patients with PDAC who received molecular testing results, 26% had actionable molecular alterations. Patients with actionable molecular alterations who received a matched therapy had a survival benefit as compared with both patients who received unmatched therapies (median overall survival (mOS) 2.58 years vs. 1.51 years, hazard ratio (HR) 0.42, 95% CI 0.26–0.68) and those who did not have an actionable molecular alteration (mOS 2.58 years vs. 1.32 years, HR 0.34, 95% CI 0.22–0.53). Thus, the adoption of biomarker driven therapies confers a substantial survival benefit to patients with PDAC. 

## 5. Maintenance Therapy

Combination chemotherapies lead to disease stabilization or even remission in a significant proportion of PDAC patients. For instance, in the ACCORD 11 trial, tumor regression and stable disease were observed in 31.6% and 38.6% of patients who received FOLFIRINOX over six months, respectively [19]. 

The next challenge for patients achieving disease control under initial combination cytostatic treatment is to minimize or avoid a worsening of side effects of the systemic therapy, in order to prevent deterioration of the quality of life. Chemotherapy holiday can be, in some situations, mandatory (i.e., for recovery from serious side effects of the systemic therapy), but a progression of the disease is in this case preprogrammed. A more reasonable concept is the de-escalation of the systemic therapy by dropping the most toxic drugs of the initial combination regimen or switching to different agents [18]. Table 3 shows an overview of current evidence for maintenance therapy in patients with advanced PDAC.

For FOLFIRINOX treated patients, withdrawal of oxaliplatin and FOLFIRI maintenance is suggested [18]. Evidence for this strategy is based on a retrospective evaluation of 22 patients treated with FOLFIRI after achieving disease control under FOLFIRINOX induction for a median of four months [38]. In this study, FOLFIRI treatment was associated with a median PFS of eight months and a favorable safety profile (18% prevalence of side effects grade 3/4). Second-line therapy with GA could be performed in 64% of patients, due to continued good performance status following tumor progression. The median OS from the beginning of FOLFIRINOX treatment in this population was 46 months. 

Another study that investigated maintenance therapy for metastatic PDA after FOLFIRINOX induction was the PRODIGE 35/PANOPTIMOX trial [39]. Notably, this trial was presented at the ASCO annual meeting in 2018 but has still not been fully published. In this prospective, randomized trial, patients were assigned to FOLFIRINOX over six months (arm A, *n* = 91), FOLFIRINOX induction for eight cycles (i.e., four months), followed by 5-FU/leucovorin maintenance therapy (arm B, *n* = 92) or alternating gemcitabine and FOLFIRI every two months (arm C, *n* = 90). Interestingly, compared to the standard arm (A), patients receiving maintenance therapy 5-FU/leucovorin (arm B) had no advantage in terms of median PFS (6.3 vs. 5.7 months) and OS (10.1 vs. 11.2 months) but a higher rate of severe neurotoxicity (grade 3/4). Conceivably, for patients receiving maintenance therapy, reescalation to FOLFIRINOX after documented progressive disease, led to a higher cumulative oxaliplatin dose with increased rate of severe neurotoxicity. The regimen used in arm C had the shortest mPFS (4.5 months) and OS (7.3 months) and deserves no further investigation. 

Capecitabine maintenance treatment was associated with a mPFS of five months in a French retrospective study on 30 patients with metastatic PDAC and disease control after four to eight cycles of FOLFIRINOX [40]. Overall, capecitabine had a favorable safety profile and the most relevant adverse event observed was hand-foot syndrome (grade 3/4 16.6%). 

Gemcitabine maintenance after GA was studied for elderly patients in a prospective observational study [41]. A total of 36 patients >70 years with metastatic or locally advanced PDAC initially received GA for a maximum of three cycles (three months). Among these patients, 86% had disease control, and thus received gemcitabine monotherapy as maintenance. Disease control rate (DCR) after six months of maintenance was 61%, median PFS was 6.4 months, and median OS was estimated at 13.4 months. Safety of gemcitabine maintenance treatment was good with grade 3 hematological toxicity in 19% of the patients.

Another interesting, non-cytostatic agent for maintenance therapy, which has been addressed in a randomized phase 2 trial for PDAC patients, was the multi-targeted receptor tyrosine kinase inhibitor, sunitinib [42]. A total of 55 patients with at least stable disease after six months of chemotherapy (mostly gemcitabine combinations) were randomized to the sunitinib or observation arm. Median PFS with sunitinib maintenance was 3.2 months as compared with 2.0 months in the calibration arm (*p* < 0.01). Although there was a slight difference of median PFS, six-months PFS was considerably higher with sunitinib (22.2% vs. 3.6%) suggesting that a subset of PDAC patients respond to sunitinib. 

In the phase 3 POLO trial, patients with metastatic PDA who had a germline BRCA1 or BRCA2 mutation and disease that had not progressed during 16 weeks of first-line platinum-based chemotherapy, were randomized to receive maintenance therapy with olaparib or placebo [28]. More than 80% of the included patients were treated with FOLFIRINOX variants before randomization. Patients receiving olaparib had a longer mPFS (7.4 months vs. 3.8 months) as compared with the placebo. However, an interim analysis at a data maturity of 46%, showed no survival benefit of olaparib over the placebo (mOS 18.9 months vs. 18.1 months).

**Table 3 cancers-12-01955-t003:** Overview of available evidence on maintenance therapy for patients with PDAC.

Authors/Study Title	Study Design	Population	Criteria for Starting Maintenance	Maintenance Strategy	mPFS	mOS	Prevalence of (Selected) Side Effects
Golan et al. 2019 [43] POLO trial	Prospective phase 3 trial, placebo-controlled, randomized 3:2	154 patients with metastatic PDAC and germline BRCA1 or BRCA2 mutation	No progressive disease after 4 months first-line platinum-based chemotherapy	Olaparib	Olaparib/placebo 7.4 vs. 3.8 months	Olaparib/placebo 18.9 vs. 18.1 months (interim analysis at a data maturity of 46%)	Serious adverse events ≥ 3° olaparib/placebo 40% vs. 23%, most common serious adverse events were anemia (11%), fatigue, or asthenia (5%) and decreased appetite (3%)
Dahan et al. 2018 [39] PRODIGE 35/PANOPTIMOX trial	Prospective phase 2 trial, randomized to 3 arms, arm A: FOLFIRINOX over 6 months, arm B: 8 cycles FOLFIRINOX followed by 5-FU/leucovorin maintenance therapy, arm C: alternating gemcitabine and FOLFIRI every 2 months	273 patients with metastatic PDAC	No progressive disease after 8 cycles FOLFIRINOX	5-FU/leucovorin	Arm A/arm B 6.3 vs. 5.7 months (arm C: 4.5 months)	Arm A/arm B 10.1 vs. 11.2 months (arm C: 7.3 months)	Neurotoxicity ≥3° arm A/arm B 10% vs. 19% (reescalation to FOLFIRINOX, higher cumulative oxaliplatin dose)
Petrioli et al. 2020 [41]	Observational prospective	31 patients with metastatic or locally advanced PDAC	No progressive disease after GA for ≤3 cycles (3 months)	Gemcitabine	6.4 months	13.4 months	Hematological toxicity 3° (19%)
Reure et al. 2016 [40]	Retrospective	30 patients with metastatic PDAC	No progressive disease after 4 to 8 cycles of FOLFIRINOX	Capecitabine	5 months (on capecitabine maintenance)	17 months	Hand-foot syndrome ≥ 3° (16.6%)
Franck et al. 2019 [38]	Retrospective	22 patients with metastatic or locally advanced PDAC	No progressive disease after FOLFIRINOX for a median of 4 months (5–12 cycles)	FOLFIRI	8 months	46 months (low data maturity)	Neutropenia and anemia ≥ 3° in both 14% of the patients

PDAC, pancreas ductal adenocarcinoma; 5-FU, 5 fluorouracil; mOS, median overall survival; mPFS, median progression free survival; FOLFIRINOX, oxaliplatin 85 mg/m2 day 1, irinotecan 180 mg/m2, leucovorin 400 mg/m2, 5-fluorouracil 400 mg/m2, 5-fluorouracil 2400 mg/m2 46 h, qd15; FOLFIRI, FOLFIRINOX (see above) with omission of oxaliplatin; GA, gemcitabine 1000 mg/m2 day 1, 8, 15, nab-paclitaxel 125 mg/m2 day 1, 8, 15, qd2; °, grade according to CTCAE v4.0.

## 6. Surgery

Many retrospective studies have shown that among patients with locally advanced, unresectable PDAC undergoing successful conversion chemotherapy and surgical exploration, those who received resection had a substantial survival benefit as compared with non-resectable patients [44,45,46]. Patient selection in these studies was based on disease control achieved with conversion therapy, whereas resectability was assessed by exploration. In the cohort with the highest survival benefit for patients receiving FOLFIRINOX conversion therapy and surgical exploration, the mOS was 37.7 months (resection) and 25.1 months (no resection), respectively [44]. 

However, in an independent cohort of patients with metastatic or locally advanced, unresectable PDAC who received FOLFIRI maintenance therapy after achieving disease control with FOLFIRINOX, the mOS was as high as 46 months [38].

The discrepancy in the mOS is substantial between patients with disease control on FOLFIRINOX therapy who undergo surgical exploration with no resection and those who receive a maintenance therapy. A plausible explanation is the chemotherapy-free interval before (usually four weeks) and after surgical exploration (at least two weeks), which is a missed opportunity for maintenance therapy.

Thus, studies on conversion chemotherapy followed by surgical resection are flawed by a missing appropriate control (i.e., maintenance therapy instead of exploration). There is no doubt that some patients can benefit from surgery, even in the metastatic stage [47,48], whereas some others can have a survival advantage by avoiding surgery. However, no convincing clinical tools or biomarkers have been identified to offer or not surgery to patients with locally advanced PDAC and recommendations are based on a “supposed” patient selection.

## 7. Treatment Options in Elderly Patients

The treatment of patients with PDAC beyond the seventh decade of life is a challenge. Indeed, they are underrepresented in prospective trials and their management is extrapolated from studies performed in younger patients, without robust evidence-based recommendations. Furthermore, therapeutic options can be limited due to impaired performance status or presence of limiting comorbidities [49]. Low-dose therapy is the standard of care for elderly patients with different gastrointestinal cancers [50,51], and evidence is also accumulating for the treatment of PDAC.

In a multicenter, randomized, open label phase 1/2 trial, the feasibility and safety of gemcitabine and nab-paclitaxel in therapy naïve elderly patients was analyzed. Succinctly, no difference between treatment in a standard schedule of three weeks on and one week off (gemcitabine 1000 mg/m^2^, nab-paclitaxel 125 mg/m^2^ day 1, 8, and 15) and gemcitabine/nab-paclitaxel biweekly (gemcitabine 1000 mg/m^2^, nab-paclitaxel 225 mg/m^2^, day 1 + 15) (B: in terms of ORR, OS, and median PFS) was observed (ORR, 20.7% vs. 22.7%; OS, 7.7 months vs. 9.8 months; and PFS, 5.4 months vs. 6.6 months). Furthermore, there were no significant differences in any of the AEs between the two dose regimens [52]. In a retrospective review, elderly patients (age ≥70) benefited from the treatment with gemcitabine/nab-paclitaxel in terms of mPFS (6.1 months) and mOS (9.3 months). However, grade ≥3 fatigue, neutropenia, and leukopenia occurred in 34.9%, 27.7%, and 25.3% of patients, respectively, and a dose reduction by at least 20% to mitigate adverse events was necessary in 84.3% of patients [53].

Guion-Dusserre and colleagues reported a retrospective review of patients >70 years with locally advanced or metastatic PDAC or colorectal cancer treated using the FOLFOX protocol (oxaliplatin 85 mg/m^2^, leucovorin 400 mg/m^2^, irinotecan 180 mg/m^2^, 5-FU 400 mg/m^2^ as bolus infusion, and 2400 mg/m^2^ over a 46 h period of infusion, every two weeks). A median of 4.5 cycles was administered to all patients. Overall, 75% of the patients initially received an adapted dose of chemotherapy, whereas a (further) dose adjustment was necessary for 26% of the patients during the course of treatment. Most common grade 3/4 hematological toxicities were diarrhea (67.3%) and neutropenia (32.7%), whereas anemia and thrombopenia were rarely represented (9.6%, and 5.8%, respectively). Moreover, in 38.5% of the patients, the therapy had to be stopped due to persistent side effects. However, with an OS of 12.5 months, patients with PDAC showed an acceptable outcome [54]. 

Conroy et al. demonstrated, in the PRODIGE 24-ACCORD trial, that adjuvant therapy with mFOLFIRINOX was superior to therapy with gemcitabine. In the group treated with mFOLFIRINOX, the dose had to be reduced in about half of the cases mainly due to hematological side effects. However, this therapy adjustment did not have a negative effect on the median OS as compared with gemcitabine (OS 54.4 months vs. 35.0 months) [23]. Although in the adjuvant setting, these data clearly show that reduced dose-intensity does not impair mFOLFIRINOX efficacy. In a small retrospective cohort analysis, 24 elderly patients >75 years were treated with a modified FOLFIRINOX protocol (most frequently used starting doses 5-FU 2400 mg/m2, irinotecan 150 mg/m^2^, and oxaliplatin 75 mg/m^2^) with acceptable median OS of 11.6 months and PFS of 3.7 months. With grade 3–4 hematological toxicities, however, treatment had to be discontinued in 42% of the patients [55]. 

In a retrospective analysis, outcomes of patients treated with FOLFIRINOX and gemcitabine/nab-paclitaxel were compared. The response rates, the OS, and PFS were comparable for patients <70 years and >70 years. However, the incidence of severe adverse events was higher in elderly patients (50% vs. 28.3%) [56]. In another retrospective analysis, the patients who received FOLFIRINOX were, on average, five years younger than those who received other chemotherapy (FOLFOX or gemcitabine). For the OS, the FOLFIRINOX protocol was significantly superior to the other two (OS 13.8 months FOLFIRINOX vs. 7.0 months FOLFOX vs. 6.7 months gemcitabine). However, severe toxicity occurred more often in the FOLFIRINOX group (especially grade 3–5 anemia and neuropathy) [57]. 

Monotherapy with gemcitabine is available as an option for patients who are too severely limited by their comorbidities, who have had treatment failure, or who have had to discontinue therapy due to toxicities. Gemcitabine monotherapy can also contribute to prolonging the OS. In contrast, Marechal and colleagues showed that especially in patients >70 years of age, the rate of side effects increased even with gemcitabine monotherapy. Patients >70 years of age had significantly more frequent grade 3/4 anemia (8.8% vs. 14.3%) and neutropenia (21.1% vs. 30.9%) as compared with patients <70 years of age [58]. The combination of gemcitabine (1000 mg/m^2^ infusion on days 1 and 8) with low dose oral capecitabine (650 mg/m^2^ twice daily on days 1 to 14) in a three-week cycle, also appeared to be a safe and effective alternative option in the elderly (ORR 19.6%, mOS 9.4 months, and grade 3–4 adverse events 33%) [59].

In conclusion, the treatment of older patients with advanced and metastatic disease remains a major challenge. Accumulating evidence suggests that low-dose treatment can preserve quality of life without compromising cancer control or survival in elderly patients amenable for tumor-specific treatment.

## 8. Staging

Staging of patients with metastatic PDAC should take into consideration imaging, tumor marker (mostly CA19-9), and patient´s general condition over time. High quality prospective studies showing the optimal timing and imaging modality, for staging patients with metastatic PDAC, are lacking. Consequently, recommendations of evidence-based guidelines [60] are not in keeping with those of clinical practice guidelines [16]. On the one hand, for example, the German S3 evidence-based guideline recommends monitoring PDAC patients by ultrasonography, if they receive systemic palliative chemotherapy outside of clinical trials, whereas no recommendations are given with respect to the time intervals to adopt. On the other hand, according to the ESMO clinical practice guidelines [16], “the efficacy of the treatment has to be evaluated every two months with a comparative CT scan. The treatment has to be stopped if a RECIST progression is observed and second-line treatment has to be discussed”. Finally, this issue is not covered by the NCCN guidelines [18]. Moreover, in countries such as Germany, the health insurance reimburses a maximum of one CT scan quarterly.

Although no evidence exists that timely tighter restaging is associated with a survival benefit, it is conceivable that performing a CT scan bimonthly has at least two advantages in the case of a progressive disease. First, to avoid unnecessary side effects of an ineffective therapy. Secondly, to offer, in a timely manner, a further line therapy, before general conditions worsen. Thus, in the patient’s best interest, we plead for guidelines and health insurances to agree in recommending a comparative standardized imaging (CT or MRI) bimonthly, with the option of extending time intervals between imaging according to the disease course.

## 9. Supportive Therapy

A thorough review of the supportive therapy in patients with metastatic PDAC is out of the scope of this review article. For this purpose, we refer to the clinical practice guidelines of the American Society of Clinical Oncology [14] and to a review article by Hendifar et al. [61] on PDAC-related weight loss. In the next paragraphs, we focus on pancreatic enzyme replacement, small bacterial overgrowth, and prophylaxis of thromboembolism.

### 9.1. Pancreatic Enzyme Replacement

Exocrine pancreatic insufficiency results from inadequate delivery of pancreatic digestive enzymes and leads to maldigestion [62].

In addition to chronic pancreatitis, the most common cause of exocrine pancreatic insufficiency is PDAC. Roughly, 70% of all cases of PDAC are localized in the pancreas head. The obstruction of the main pancreatic duct associated with PDAC of the pancreas head prevents the pancreatic enzymes from reaching the duodenum [63]. 

Exocrine pancreas insufficiency has been reported in 50–90% of patients with unresectable PDAC [64]. Weight loss, sarcopenia, and reduced ability to tolerate chemotherapy or surgery are all possible consequences of exocrine pancreas insufficiency [65,66]. 

Diagnosis of exocrine pancreatic insufficiency in a patient with PDAC is mainly based on clinical judgement (steatorrhea, weight loss, abdominal discomfort, and bloating), whereas the measurement of fecal elastase-1 lacks sensitivity and specificity [67]. Thus, in a patient with PDAC and clinically suspected exocrine pancreatic insufficiency, empiric therapy with pancreatic enzyme replacement therapy (PERT) should be started without formal testing [60].

The normal pancreas produces roughly 900,000 United States Pharmacopeia (USP) units of lipase with an average meal. It has been estimated that 10% of this amount (i.e., 40,000–50,000 USP units with each meal) are sufficient to correct steatorrhea [68].

Several European studies have reported that many patients receive far less PERT or do not receive PERT at all [69,70]. Recently a retrospective study covering over 80 U.S. healthcare patients showed that only one in five patients with PDAC was prescribed PERT, whereas a minimally effective PERT dosage was prescribed in less than 10% [71].

### 9.2. Small Bacterial Overgrowth

Small intestinal bacterial overgrowth (SIBO) is a clinical condition, characterized by exceeding 10^5^–10^6^ bacterial organisms/ml in the small intestine. Patients with SIBO suffer from symptoms such as bloating, chronic diarrhea, weight loss, malabsorption, and can develop nutritional deficiencies, osteoporosis, and anemia [72].

A number of clinical conditions, including reduced gastric acid secretion, exocrine pancreatic insufficiency, small intestine dysmotility, disturbances in the gut immune function, and anatomical abnormalities of the gastrointestinal tract are associated with an increased risk of developing SIBO [73]. Patients with PDAC are also prone to SIBO. In a prospective study, the prevalence of SIBO among 30 patients with PDAC was 63.3% [74]. Moreover, 25–50% of the patients with exocrine pancreatic insufficiency, a condition common among patients with PDAC, had concomitant SIBO that could account for the persistence of diarrhea despite adequate pancreatic enzyme supplementation [75]. Although the effect of SIBO eradication has not been prospectively studied in this specific subgroup of patients, SIBO eradication is likely to also improve symptoms and quality of life of patients with PDAC. Accordingly, in a case report of a 62-year-old Caucasian male with advanced PDAC, diarrhea greatly improved following SIBO therapy [76]. 

### 9.3. Prophylaxis of Thromboembolism

Roughly, 23–36% of patient with PDAC develop a venous thromboembolism (VTE) [77] and this has been associated with interruption in treatment and increased mortality (HR 2.6, 95% CI 2.3–2.8) [78]. Tissue factor (TF)-positive microvesicles could play a role in the pathophysiology of VTE among cancer patients. Interestingly, the highest levels of TF-positive microvesicles have been identified in PDAC and this correlated directly with the risk of VTE [79,80,81].

In the multicenter phase 3b CASSINI study, ambulatory cancer patients at high risk of VTE (Khorana score ≥ 2) were randomized to receive rivaroxaban 10 mg/d or placebo. Prophylactic rivaroxaban was associated with a reduction in the incidence of thromboembolism (HR 0.40, 95% CI 0.20–0.80) as compared with the placebo [82]. The risk of VTE among patients with advanced PDAC was higher as compared with patients with primary cancer localized elsewhere. In a prespecified subgroup analysis of the CASSINI study, in patients with advanced PDAC, the therapy with rivaroxaban was associated with a reduced risk of VTE without an increase in major bleeding events (HR 0.35, 95% CI 0.13–0.97, *p* = 0.03), suggesting a potential benefit of prophylactic rivaroxaban therapy in this high-risk group [83].

### 9.4. Improving Care in Patients with PDAC

The traditional indicators of effectiveness in oncology (i.e., PFS, OS, etc.) do not always mirror patients’ perspective. A more thorough picture of the effectiveness of a tumor therapy can be achieved by incorporating patient-reported outcomes (PROs) in clinical trials. PRO data are usually collected via questionnaires filled out by the patient in scheduled intervals. In the near future, due to further development of telemedicine and wearable devices, uncomplicated, but closer monitoring of a patient’s symptoms in clinical trials (and daily routine) should be possible [84,85]. This could help evaluate a patient’s experience more precisely, thus, allowing, in the future, a more patient-centered treatment assessment. Furthermore, telemedicine and wearable devices could help to identify PDAC patients that need a closer follow-up and, conversely, those who could benefit from a reduced frequency of visits to the oncology department.

## 10. Conclusions

Advances in systemic therapy for patients with advanced PDAC are limited, but biomarker driven therapies following molecular profiling have the potential to improve survival among the subgroup of patients with actionable molecular alterations. 

A new subtyping system has identified two distinct PDAC subtypes with significant implications for prognosis and treatment response to FOLIFIRNOX. One of the next tasks is to develop a subtyping system for selecting patients with locally advanced disease that could benefit from conversion therapy and surgery.

In the elderly, low-dose chemotherapy can offer a survival benefit without compromising quality of life. 

Maintenance therapy for PDAC is taking hold in clinical practice. High-quality evidence is limited to the POLO data on olaparib for the selected subgroup of patients carrying a germline BRCA1 or BRCA2 mutation, and thus prospective trials in this area are warranted. 

## Figures and Tables

**Table 1 cancers-12-01955-t001:** Pancreatic ductal adenocarcinoma (PDAC) subtypes, according to Bailey et al. [10].

Subtype	Characteristics	Incidence	Survival (Months)
Squamous	Enriched for activated α6β1 and α6β4 integrin signaling and activated EGF signaling. Hypermethylation of genes that govern pancreatic endodermal cell-fate determination leading to a complete loss of endodermal identity.	31%	13.3
Aberrantly differentiated endocrine exocrine (ADEX)	Transcriptional networks in later stages of pancreatic development and differentiation. Transcription factors NR5A2, MIST1 important in acinar cell differentiation and pancreatitis regeneration. Genes associated with endocrine differentiation and MODY.	19%	25.6
Pancreatic progenitor	Pancreas development, gene programs regulating fatty acid oxidation, steroid hormone biosynthesis, drug metabolism, and O-linked glycosylation of mucins.	29%	23.7
Immunogenic	Immune infiltrate, B cell signaling pathways, antigen presentation, CD4+ T cell, CD8+ T cell, and Toll-like receptor signaling pathways.	21%	30

**Table 2 cancers-12-01955-t002:** Definition of FOLFIRINOX and modified (m) FOLFIRINOX according to Conroy et al. [19,23].

Regimen	Oxaliplatin *	Leucovorin *	Irinotecan #	Fluorouracil ^a^	Fluorouracil ^b^	Selected Adverse Events ≥ 3°
FOLFIRINOX [19]	85 mg/m^2^	400 mg/m^2^	180 mg/m^2^	400 mg/m^2^	2.400 mg/m^2^	Sensory neuropathy 9.3%, fatigue 23.6%, neutropenia 45,7% (filgrastim was administered in 43% of the patients), vomiting 14.5%, diarrhea 12.7%
mFOLFIRINOX [23]	85 mg/m^2^	400 mg/m^2^	180 mg/m^2^ (after the enrollment of 162 of 493 patients, the dose of irinotecan was reduced to 150 mg/m^2^)	No fluorouracil bolus was administered	2.400 mg/m^2^	Sensory neuropathy 9.0%, fatigue 11.0%, neutropenia 28.4% (filgrastim was administered in 62% of the patients), vomiting 5.1%, diarrhea 18,6% (20.0% who received irinotecan at a dose of more than 175 mg/m^2^ and 17.1% who received a dose of 150 mg/m^2^)

Dose reduction can be considered in elderly and fragile patients, as well as to prevent recurrence of side effects. * 2-h intravenous infusion; # 90-min intravenous infusion; ^a^, intravenous bolus; ^b^, infusion over a 46-h period; °, grade according to CTCAE v4.0.

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
