# Peer review of "Advanced Pancreatic Ductal Adenocarcinoma: Moving Forward"

_cancers, 2020, doi:10.3390/cancers12071955_

Round 1

Reviewer 1 Report

This is an excellent review article. With increasing incidence of PDAC, the article is timely and comprehensive. I did not find any major issues.

One minor issue:

The prognosis of PDAC where the precursor is IPMN (either side branch or main duct or mixed) is better than PanIN pathway. Authors could consider commenting on outcomes of PDAC and treatment when known precursor is IPMN (rather than PanIN pathway). This has importance since incidental cysts are increasingly detected more so now than ever before. 

Author Response

Reviewer 1

This is an excellent review article. With increasing incidence of PDAC, the article is timely and comprehensive. I did not find any major issues.

-->Thank you.

One minor issue:

The prognosis of PDAC where the precursor is IPMN (either side branch or main duct or mixed) is better than PanIN pathway. Authors could consider commenting on outcomes of PDAC and treatment when known precursor is IPMN (rather than PanIN pathway). This has importance since incidental cysts are increasingly detected more so now than ever before.

--> We agree with Reviewer 1,. The following sentence has been added to the updated version of the article, in the Introduction section: “For PDAC arising within intraductal papillary mucinous neoplasm (IPMN) the prognosis is more favorable with a 5-year survival reported between 40% and 60%. However, this is due to comparatively lower tumor stages and better pathologic characteristics at time of surgery. For advanced IPMN-associated PDAC the prognosis is comparable to non-IPMN-associated PDAC”.

Reviewer 2 Report

The manuscript “Advanced pancreatic ductal adenocarcinoma: moving forward” is in general a well-organized, scholarly, and useful review of the current state of advanced pancreatic cancer treatment.  What is mis-leading is the inclusion of ‘moving forward’ as part of the title.  There is little evidence that this review includes an informed speculation on where the field might be heading. 

Specific comments: 

I.  The title should either drop the implication that there is a forward-looking component of the review or incorporate a final paragraph that is clearly speculative but gives some insight and encouragement to those interested and caring for patients in this field.   Two possible examples of topics that would fit this need:

          A.  There are sections on therapies indicated by germline BRCA1/2/PALB2 and dMMR/MSI-H molecular alterations, but a broadening of this concept to incorporate the reports that approximately 25% of PDAC cases have molecular alterations that indicate a targeted therapy could indicate that the future may hold a expansion of molecularly-driven approaches for this disease.  The incorporation of approvals for tumor agnostic NTRK targeted therapies and results from real-world studies such as Lancet Oncology 21:508 (2020) would support this as the basis for future studies and treatment paradigms. 

          B.   The incorporation of supportive care as a critical component of the care of advanced pancreatic cancer patients is a very welcome addition to this review but it is a missed opportunity not to speculate on what the future could hold in this area.  Improved understanding of the underlying causes of malabsorption and weight loss, and the possibility of using patient-reported outcomes or results from wearable devices to improve the monitoring and care of patients are the types of things that would really benefit this patient population.  A review found in The Oncologist 23, 1-11, 2018 identifies three distinct causes of weight-loss with distinct interventions that might be a useful framework for envisioning the future for supportive care of advanced PDAC.

II.  Section 2 is a nice analysis of the Bailey et al PDAC subtypes.  The authors may not be aware of a recent paper in which subtypes by three different groups were harmonized to reveal two distinct subtypes with therapeutic implications (Clin Cancer Res 26: 82-92, 2020).  This is a simpler classifier with some clinical correlative data that could be included to update the review.

III.  It seems unusual not to mention pain control as part of the Supportive care for advanced PDAC.

Author Response

Reviewer 2

Comments and Suggestions for Authors

The manuscript “Advanced pancreatic ductal adenocarcinoma: moving forward” is in general a well-organized, scholarly, and useful review of the current state of advanced pancreatic cancer treatment.  What is misleading is the inclusion of ‘moving forward’ as part of the title.  There is little evidence that this review includes an informed speculation on where the field might be heading.

à We implemented our review article to overcome this issue, as advised by Reviewer 2 and 3.

Specific comments:

  1. The title should either drop the implication that there is a forward-looking component of the review or incorporate a final paragraph that is clearly speculative but gives some insight and encouragement to those interested and caring for patients in this field. Two possible examples of topics that would fit this need:

à We give some insight and encouragement to those interested and caring for patients with PDAC in the conclusion section.

  1. There are sections on therapies indicated by germline BRCA1/2/PALB2 and dMMR/MSI-H molecular alterations, but a broadening of this concept to incorporate the reports that approximately 25% of PDAC cases have molecular alterations that indicate a targeted therapy could indicate that the future may hold a expansion of molecularly-driven approaches for this disease. The incorporation of approvals for tumor agnostic NTRK targeted therapies and results from real-world studies such as Lancet Oncology 21:508 (2020) would support this as the basis for future studies and treatment paradigms.

à According to the suggestioins of Reviewer 2 we divided the section “Mainstay” in 3 subsections with first line treatments, second line treatments and biomarker driven treatments including data on NTRK and from the real-world study published on Lancet Oncology.This last real-world study is also included in the conclusion section to encourage collegues caring for patients with PDAC.

  1. The incorporation of supportive care as a critical component of the care of advanced pancreatic cancer patients is a very welcome addition to this review but it is a missed opportunity not to speculate on what the future could hold in this area. Improved understanding of the underlying causes of malabsorption and weight loss, and the possibility of using patient-reported outcomes or results from wearable devices to improve the monitoring and care of patients are the types of things that would really benefit this patient population. A review found in The Oncologist 23, 1-11, 2018 identifies three distinct causes of weight-loss with distinct interventions that might be a useful framework for envisioning the future for supportive care of advanced PDAC.

à We thank reviewer 2 for pointing on the importance of weight loss in PDAC patients. A thorough review on weight loss in patients with advanced PDAC was out of the scope of this review article, also due to space constraints. However, we think that the mentioned review by Hendifar et al. is critical for understanding weight loss in PDAC patients and added a comment with citation to the updated version of the manuscript.

We added a new paragraph on patient-reported outcomes and results from wearable devices.

  1. Section 2 is a nice analysis of the Bailey et al PDAC subtypes. The authors may not be aware of a recent paper in which subtypes by three different groups were harmonized to reveal two distinct subtypes with therapeutic implications (Clin Cancer Res 26: 82-92, 2020). This is a simpler classifier with some clinical correlative data that could be included to update the review.

à Thank you for your advice. The article by Rashid et al has been cited and discussed in the updated version of the article. Morover, as a “Moving forward point”, this new subtype classification is included in the Conclusion section.

III. It seems unusual not to mention pain control as part of the Supportive care for advanced PDAC.

à We agree that pain management is a very important issue for patients with advanced PDAC. A thorough review of supportive therapy in patients with advanced non-resectable or metastatic PDAC was out of the scope of this review article, also for space constraints. For this purpose, we refer in the current version of the manuscript to the clinical practice guidelines of the American Society of Clinical Oncology. We decided to focus on pancreatic enzyme replacement, small bacterial overgrowth and prophylaxis of thromboembolism. We specify this aspect in the updated version of the manuscript.

Reviewer 3 Report

Franck C et al. successfully reviewed clinical role of systemic therapy in patients with unresectable (UR) pancreatic ductal adenocarcinoma (PDAC) with the new knowledge of Olaparib in patients with germline BRCA1/2 mutations. Moreover, they revealed the clinical evidence of maintenance chemotherapy, and supportive therapy in addition to the role of small bacterial overgrowth and prophylaxis of thromboembolis.

The manuscript is well-written and easy to read. It provided the well-summarized information for the next step to the readers.

I have two minor comments.

  1. One of the next steps for improving long-term survival in patients with UR-PDAC is to select patients for conversion surgery even in initially UR-PDAC. Several articles revealed the improved prognosis in these patients. This treatment option can be one of “moving forward” in this area. I hope the authors add the paragraph of conversion surgery or additional surgery during multimodal treatments.
  • Hackert T, et al. Ann Surg. 2016;264:457–63.
  • Satoi S et al. J Hepatobiliary Pancreat Sci. 2013;20:590–600.
  • Wright GP, et al. J Gastrointest Surg. 2016 ;20:1830-1835.
  • Michelakos T, et al. Ann Surg. 2019;269:733-40.
  • Rangelova E, et al. Ann Surg. 2019 Apr 2. doi: 10.1097/SLA.0000000000003301.
  • Satoi S, et al. Ann Gastroenterol Surg. 2019;4:6-13. 
  1. “8. Materials and Methods” should be moved to the next to the Introduction.

I appreciate for giving me an opportunity to review this interesting manuscript.

Author Response

Reviewer 3

Comments and Suggestions for Authors

Franck C et al. successfully reviewed clinical role of systemic therapy in patients with unresectable (UR) pancreatic ductal adenocarcinoma (PDAC) with the new knowledge of Olaparib in patients with germline BRCA1/2 mutations. Moreover, they revealed the clinical evidence of maintenance chemotherapy, and supportive therapy in addition to the role of small bacterial overgrowth and prophylaxis of thromboembolis.

The manuscript is well-written and easy to read. It provided the well-summarized information for the next step to the readers.

à Thank you.

I have two minor comments.

One of the next steps for improving long-term survival in patients with UR-PDAC is to select patients for conversion surgery even in initially UR-PDAC. Several articles revealed the improved prognosis in these patients. This treatment option can be one of “moving forward” in this area. I hope the authors add the paragraph of conversion surgery or additional surgery during multimodal treatments.

Hackert T, et al. Ann Surg. 2016;264:457–63.

Satoi S et al. J Hepatobiliary Pancreat Sci. 2013;20:590–600.

Wright GP, et al. J Gastrointest Surg. 2016 ;20:1830-1835.

Michelakos T, et al. Ann Surg. 2019;269:733-40.

Rangelova E, et al. Ann Surg. 2019 Apr 2. doi: 10.1097/SLA.0000000000003301.

Satoi S, et al. Ann Gastroenterol Surg. 2019;4:6-13.

à We thank Reviewer 3 for this constructive comment. We added an extra paragraph on surgery and speculate in the Conclusion section on a possible subtyping system able to select patients with advanced disease that may benefit from conversion surgery.

“8. Materials and Methods” should be moved to the next to the Introduction.

à The section “Materials and Methods” follows now the introduction. (“2. Materials and Methods”).